# An Efficient 5-Aminolevulinic Acid Photodynamic Therapy Treatment for Human Hepatocellular Carcinoma

**DOI:** 10.3390/ijms241310426

**Published:** 2023-06-21

**Authors:** Abhishek Kumar, Florian Pecquenard, Martha Baydoun, Alexandre Quilbé, Olivier Moralès, Bertrand Leroux, Lynda Aoudjehane, Filomena Conti, Emmanuel Boleslawski, Nadira Delhem

**Affiliations:** 1Univ. Lille, Inserm, CHU Lille, U1189-ONCO-THAI-Assisted Laser Therapy and Immunotherapy for Oncology, F-59000 Lille, France; asuknoit@gmail.com (A.K.); florian.pecquenard@hotmail.fr (F.P.); martha.e.b@hotmail.com (M.B.); aquilbe@gmail.com (A.Q.); olivier.morales@ibl.cnrs.fr (O.M.); bertrand.leroux@inserm.fr (B.L.); 2CHU Lille, Service de Chirurgie Digestive et Transplantations, Université de Lille, F-59037 Lille, France; 3Univ. Lille, CNRS, Inserm, CHU Lille, UMR9020-U1277-CANTHER–Cancer Heterogeneity Plasticity and Resistance to Therapies, F-59000 Lille, France; 4INSERM, Institute of Cardiometabolism and Nutrition (ICAN), Sorbonne Université, F-75013 Paris, France; lynda.aoudjehane@inserm.fr (L.A.); filomena.conti@aphp.fr (F.C.); 5INSERM, Centre de Recherche Saint-Antoine (CRSA), Sorbonne Université, F-75012 Paris, France; 6Assistance Publique-Hôpitaux de Paris (AP-HP), Pitié–Salpêtrière Hospital, Department of Medical Liver Transplantation, F-75013 Paris, France

**Keywords:** photodynamic therapy, hepatocellular carcinoma, 5-ALA, anti-tumor immunotherapy, anti-tumor immunity

## Abstract

Photodynamic therapy (PDT) is a two-stage treatment relying on cytotoxicity induced by photoexcitation of a nontoxic dye, called photosensitizer (PS). Using 5-aminolevulinic acid (5-ALA), the pro-drug of PS protoporphyrin IX, we investigated the impact of PDT on hepatocellular carcinoma (HCC). Optimal 5-ALA PDT dose was determined on three HCC cell lines by analyzing cell death after treatment with varying doses. HCC-patient-derived tumor hepatocytes and healthy donor liver myofibroblasts were treated with optimal 5-ALA PDT doses. The proliferation of cancer cells and healthy donor immune cells cultured with 5-ALA-PDT-treated conditioned media was analyzed. Finally, therapy efficacy on humanized SCID mice model of HCC was investigated. 5-ALA PDT induced a dose-dependent decrease in viability, with an up-to-four-fold reduction in viability of patient tumor hepatocytes. The 5-ALA PDT treated conditioned media induced immune cell clonal expansion. 5-ALA PDT has no impact on myofibroblasts in terms of viability, while their activation decreased cancer cell proliferation and reduced the tumor growth rate of the in vivo model. For the first time, 5-ALA PDT has been validated on primary patient tumor hepatocytes and donor healthy liver myofibroblasts. 5-ALA PDT may be an effective anti-HCC therapy, which might induce an anti-tumor immune response.

## 1. Introduction

Hepatocellular carcinoma (HCC) is the most common primary liver cancer and has the second-highest cancer-associated mortality worldwide [1]. Generally arising from chronic liver cirrhosis, i.e., a recurrent cycle of cell death and regeneration in the liver, HCC exhibits a highly complex tumor microenvironment (TME) with hypervascularization [2,3]. Despite advances in diagnostic, locoregional, and surgical therapies, complete HCC treatment has remained a challenge [4,5,6]. The inability of available therapies to induce an effective anti-cancer immune response causes tumor recurrence in several cases. Therefore, this field seeks the development of novel therapies, which can not only induce tumor cell death but also activate anti-tumor immunity, with support for tumor visualization before and during surgery and later in prognosis.

Photodynamic therapy (PDT) is a minimally invasive treatment that uses a light-sensitive drug called a photosensitizer (PS). Following the accumulation of the PS in the tumor, it is excited by light of an appropriate wavelength, which results in the production of reactive oxygen species (ROS) capable of inducing cytotoxicity [7]. This two-step treatment process makes the therapy highly specific thereby giving localized destruction of the tumor with minimal side effects [8]. Additionally, the fluorescence property of the PS could aid in tumor visualization and diagnosis [9,10]. The identification of small tumors is of high importance for the resection and improvement of patient outcomes. Its ability to induce an anti-tumor immunity through the induction of immunogenic cell death has been demonstrated in some key studies [11,12]. One of the most widely studied PS is protoporphyrin IX (PpIX), which is metabolized from the pro-drug 5-aminolevulinic acid (5-ALA) using the endogenous heme-biosynthetic pathway. 5-ALA-mediated PDT has become a standard in dermatology for skin disorders like actinic keratoses, along with other malignant and premalignant diseases [13,14]. Clinical trials for several superficial and cutaneous cancer types are underway [15].

Here, we investigated the use of 5-ALA-mediated PDT as a potential anti-HCC therapy. Our study aimed to use 5-ALA PDT to induce in vitro tumor cell death in HCC cell lines and study its impact on the immune system. We validated our findings by evaluating the impact of 5-ALA PDT on primary tumor hepatocytes from hepatectomy specimens in patients with HCC. We also studied the impact of the therapy on primary healthy donor liver myofibroblasts, in terms of their cellular viability and activation state, to observe possible fibrosis induced by the therapy. Subsequently, we supported our study by evaluating the impact of 5-ALA PDT on a subcutaneous SCID mice model of HCC. Our results will help the future development and application of PDT as an effective treatment for HCC and the evolution of a better treatment strategy.

## 2. Results

### 2.1. PpIX Accumulation in HCC Cell Lines

In order to validate the cellular metabolism of 5-ALA into PpIX, which is the effective PS mediating PDT-activated tumor cell death, we observed the intracellular accumulation of PpIX in the three HCC cell lines. Figure 1 demonstrates the fluorimetry-based intracellular accumulation of PpIX in the three HCC cell lines when cultured with 0.6 mM of 5-ALA for varying durations. We observed a continuous increase in the levels of PpIX fluorescence at the earliest of 1 h of incubation for the three cell lines. The intracellular PpIX fluorescence levels continued increasing until 18 h, after which the levels saturated for all the cell lines. Interestingly, we also observed an extracellular increase in the PpIX fluorescence levels from roughly 4 h, which continued increasing until 24 h. From this, we can conclude that the cells maintain a constant metabolism of the exogenous 5-ALA until 18 h, after which they became saturated, and thus the cells started to secrete PpIX. An equilibrium between PpIX synthesis and secretion might have been instantaneously established. We did not observe any change in the intracellular or extracellular PpIX fluorescence levels of the Non-Treated control. The Intracellular PpIX fluorescence levels for the three cell lines are similar.

### 2.2. In Vitro Phototoxicity in HCC Cell Lines

Next, we established an optimal 5-ALA PDT dose for each of the cell lines. When subjected to varying concentrations of 5-ALA with varying illumination doses, we observed a dose-dependent effect on the cellular viability of these cell lines, 24 h post-illumination (Figure 2). The cellular viability started to decrease with 0.1 mM of 5-ALA and 0.6 J/cm^2^ of illumination dose for HuH7, and 0.2 mM of 5-ALA and 1.8 J/cm^2^ of illumination dose for Hep3B and HepG2. Despite slight variations among different doses, we did not observe any significant change in the normalized viability of the cell lines for the Non-Treated or 5-ALA controls. We observed a significant increase in the viability of cells treated with light only. This effect can be due to the photoactivation of the cells, i.e., the stimulation of cells by light due to endogenous photosensitizers of the cells, the reason for which is still undocumented.

From this dose-dependent effect, we obtained an IC50 dose (half-maximal inhibitory concentration) of 0.6 mM of 5-ALA with 0.6 J/cm^2^ of illumination dose for HuH7 and 0.6 mM of 5-ALA with 1.8 J/cm^2^ of illumination dose for Hep3B and HepG2. We further investigated the impact of 5-ALA PDT on cell proliferation and cytotoxicity. Figure 3 shows a significant decrease in cell proliferation of all the cell lines treated with their respective IC50 dose of 5-ALA PDT. While all the cell lines showed a significant decrease when treated with 5-ALA only, along with a decrease in the Light Only control of HuH7, it is the combined impact of 5-ALA and the illumination dose that exhibits the most significant decrease for all the cell lines.

Previous studies have shown that HuH7 cells have an over-expression of the transcription factor p53, while Hep3B are p53-null and HepG2 p53-wild-type [16]. Taking into consideration the varying sensitivity of the three HCC cell lines to 5-ALA PDT, the difference in their p53 state, and the central role of p53 in cell death regulation, we hypothesized that there can be p53-dependent or -independent cell death pathways involved. Hence, we investigated the type of cell death induced by 5-ALA PDT on the most sensitive cell line, HuH7. Figure 4 highlights the percentage of HuH7 cells that are viable, necrotic, or apoptotic after treatment with an optimal dose of 5-ALA PDT at different time points post-illumination. Starting from 1 h post-illumination, we did not observe any significant change in the levels of the population for all the control conditions. However, at 4 h post illumination, we observed a significant increase in the necrotic cell population treated with 5-ALA PDT, which rose to roughly three-fold at 24 h post-illumination with respect to the Non-Treated condition. The positive controls of the heat treatment and staurosporine treatment showed a significant increase in necrotic and apoptotic populations, respectively, with reference to the Non-Treated condition. Our results highlight that 5-ALA PDT causes the necrosis of HuH7 cells, thereby highlighting a p53-mediated necrotic cell death pathway.

When analyzed for cytotoxicity (Figure 5) via LDH release, we observed a significant increase of approximately 80–90% of PDT-treated cells at 24 h post-illumination. There was also an increase of 60–80% cytotoxicity in cells in the Light Only control. 5-ALA has no impact on the cytotoxicity of the cells. Under the microscope, PDT treatment with 5-ALA induced morphological changes, with all three cell lines taking on a rounded shape and becoming brighter. It should be noted that the control conditions, 5-ALA and illumination, only have an impact on mitochondrial metabolism, cell proliferation and LDH release. Since we did not observe any significant change in the necrotic or apoptotic population of the cells subjected to these control conditions, we believe the control conditions might affect the proliferation or mitochondrial metabolism but do not induce cell death and that further investigation is required regarding the impact of 5-ALA and illumination on these cellular aspects.

Since photodynamic therapy relies on both PS and light-dose properties, the evaluation of the best-combined conditions for each parameter implies a difference in the results themselves. We used three different cell lines with different specificities, notably according to the p53 status. In fact, HuH7 cells over-express p53, which goes hand in hand with their hypersensitivity to PDT, compared to HepG2, which do not express it, or to Hep3B, which are considered wild types for p53 expression, partly explaining the lower sensitivity recorded for them.

Then, only considering the photophyisical and biochemical part of PDT, one can assume, as a principle, that a minimum amount of energy (coming from photons) has to be delivered to the PS from the light to have the photoactivated PS produce enough reactive oxygen species. Indeed, according to the Perrin–Jablonski diagram illustrating the mechanisms of PDT, including the type I and type II photoreactions, PSs absorb photons of light and cause excitation to the singlet excited state. The singlet-excited PS can decay to the ground state, releasing energy in the form of fluorescence, a principle used for fluorescence-based experiments (Figure 1). It is possible that the singlet is converted to a long-lived triplet excited state, capable of transferring energy to another triplet (ground state of oxygen) or transferring electrons to oxygen, producing a series of ROS or superoxides. Moreover, even though there is sufficient energy to get obtain ROS from the PS, the internal concentration of the PS needs also to be sufficient to allow enough ROS formation to counteract the cell resistance and degrade the cell components triggering cell death.

As a consequence, the accumulation of both the minimum of required photophysical and biochemical triggers required, combined with the internal biological properties of the different cell lines, largely explain the way they respond to PDT and the variations observed in the experimental parameters required IC50 values for all of them.

All in all, these results confirm that 5-ALA PDT induces cell death in the HCC cell lines.

### 2.3. Impact of 5-ALA PDT Treated Conditioned Media on HCC Cell Lines

It is widely known that cancer actively communicates with its microenvironment by secreting factors such as exosomes, cytokines, and chemokines. Known as tumor secretoma they is one of the key components of the TME that can effectively limit the overall efficacy of an anti-tumor therapy via immune suppression, resistance to the therapy, cancer stem cell induction, or future tumor recurrence. Hence, we studied the impact of 5-ALA-PDT-treated secretoma, represented by conditioned media harvested from cancer cell lines with or without 5-ALA PDT treatment. The overall objective here is to investigate how the 5-ALA PDT treatment will impact the tumor cells that were not eliminated by the treatment due to limited drug or light penetration.

We used 5-ALA-PDT-treated conditioned media harvested 24 h post-illumination to culture respective cancer cell lines and observed their viability and proliferation over a course of 5 days of culture. Our results in Figure 6a show a significant two-to-four-fold decrease in normalized proliferation of these cancer cells which were 5-ALA-PDT-treated or 5-ALA Only (control) conditioned media, with no significant difference between them. These results once again highlight a possible cytotoxic effect of 5-ALA on HCC cells. Figure 6b shows a slightly significant change in normalized viability for the various conditions across the cell lines. These data reflect that 5-ALA or its intermediates are responsible for the secretion of certain unknown factors by the cancer cells, which are responsible for the cytostatic effect on other cancer cells, via an autocrine or a paracrine signaling process.

### 2.4. Impact of 5-ALA PDT on Healthy Donor Human Myofibroblasts

In order to investigate whether 5-ALA-mediated PDT could induce fibrosis, primary human liver myofibroblasts from three different healthy donors were subjected to two doses of 5-ALA PDT to analyze their viability and activation state 24 h post-illumination. As shown in Figure 7a, despite slight variations, we observed no statistically significant decrease in cellular viability for the three donors with respect to the Non-Treated control. Please note that the extreme variability in the results here is due to reduced metabolic activity of the primary fibroblasts upon subsequent culture.

The proliferation of fibroblast cells is a hallmark of fibrosis. We then tested the ability of the myofibroblasts to proliferate under the different control conditions and PDT. As shown in Figure 7b, the presence of 5-ALA decreases the proliferation ability of the cells, but more importantly, under PDT, and whatever the light dose is used to trigger it, PDT significantly reduces myofibroblast proliferation, indicating the favorable role of PDT in the control of fibrosis. To clarify this, we further examined fibrosis induction in two different aspects. The first one being the activation of several genes associated with fibrosis, and the second one consisting of the quantification of collagen I secretions, another hallmark of fibrosis. Despite a decrease in the fold change of the mRNA transcripts of the genes, we did not observe any significant increase for all the genes tested (Figure 8a). However, we observed a slight significant decrease in collagen I secretion of approximately 100 ng/mL for both the PDT doses (shown in Figure 8b). All these results highlight that there might be slight PDT- or 5-ALA-associated toxicity in healthy donor fibroblasts, but no activation or fibrosis is induced.

### 2.5. Impact of 5-ALA-PDT-Treated Conditioned Media on Human PBMCs

As described previously, tumor secretoma after PDT can directly influence cancer cell metabolism, and as a part of the TME, it can also interact with other intra- and peritumor components, such as the tumor-infiltrating immune cells. This interaction is of utmost importance regarding the immune control or tolerance of the tumor, which thereby leads to further anti- or protumor immune response, respectively [17].

To assess the impact of the secretomes from the HCC cell lines on the proliferation of human PBMCs, we stimulated human PBMCs from healthy donors with PHA and then cultured them with the conditioned media of the cell lines from different conditions. Figure 9a shows that the 5-ALA-PDT-treated conditioned media could significantly increase the normalized proliferation of human PBMCs starting from day 2 of culture for all three cell lines. This increased proliferation peaks at day 3 by two-to-three fold of the Non-Treated condition, after which it wears off. We observed minor but significant changes in the normalized viability levels of the different groups of the cell lines (Figure 9b). These results highlight clonal expansion, where a certain activated immune population rapidly proliferates when encountering an antigen. Such clonal expansion leads to antigen-specific immune response activation.

### 2.6. Preclinical Assessment of the 5-ALA PDT Efficiency in a SCID Mice Model of HCC

Based on the preceding findings, we validated our results in vivo. To achieve this, we developed a human HCC xenograft model using SCID mice for the preclinical assessment of 5-ALA PDT. We subcutaneously injected the tumorigenic cell line in the SCID mice, and after a tumor of an appropriate volume was obtained, we injected doses of 5-ALA PDT using a previously established protocol (unpublished study). As reported in Figure 10a, we demonstrated that the mice group treated with 5-ALA PDT had a decrease in tumor growth rate. We also observed a significant decrease in average normalized bioluminescence with respect to the Non-Treated group. These results, therefore, suggest that even though the tumor is not completely eliminated by 5-ALA PDT, we observe a clear decrease in the tumor growth rate. A point to note here is that in order to maintain a clinical set-up, we utilized tumors of a large volume, equivalent (on a mouse scale) to large HCC nodules in patients subjected to hepatectomy. We are aware that a tumour of this size cannot be eliminated by 5-ALA PDT alone, but it does limit the rate of tumour growth. This observation can be further augmented by the activation of the anti-tumor immune response by 5-ALA PDT, which has been demonstrated through in vitro studies.

In order to analyze 5-ALA, illumination, or 5-ALA PDT biosafety, we observed the changes in the weight of the mice during the course of the treatment. As highlighted in Figure 10b, we observed no change in the weight of the mice until they were sacrificed, suggesting the biosafety of the treatment. These results also confirm that there are no 5-ALA- or light-associated toxicities that were observed during in vitro studies.

### 2.7. Impact of 5-ALA on HCC Patient Tumor Hepatocytes

In order to validate our findings on cell lines and the in vivo model, the two doses of 5-ALA PDT mentioned above were tested on primary tumor hepatocytes. Tumor hepatocytes harvested from four different HCC patient tumors were subjected to two different doses of 5-ALA PDT at 0.6 J/cm^2^ and 1.8 J/cm^2^ of illumination dose with 0.6 mM of 5-ALA to analyze their viability at different time points post-illumination. Our results (Figure 11) demonstrate that at 3 days post-illumination, there is a significant decrease in the viability of the tumor hepatocytes treated with both doses of 5-ALA PDT, along with those treated in the 5-ALA Only control. The viability continued to decrease until 10 days post-illumination. No viability decrease was observed for the Light Only control group with respect to the Non-Treated condition. This calls to attention that 5-ALA or its intermediates themselves are causing toxicity in the tumor hepatocytes and not the combined effect, which needs to be further investigated.

## 3. Discussion

In the present study, we highlighted the impact of 5-ALA PDT on HCC, where we tested the therapy on respective cell lines, as well as primary HCC patient samples and donor healthy liver myofibroblasts, followed by a SCID mice model for the same. We also studied the impact of the secretomes on human immune cells and cancer cells. It should be noteworthy that PpIX selectively accumulates in cancerous cells rather than in healthy cells as it is the precursor for heme via the ferrochelatase reaction, which is the rate-limiting enzyme in the whole pathway with reduced expression and/or activity in the cancer cells [18,19]. The photophysical properties of PpIX are already very well established and summed up [20], therefore the UV-visible absorption, fluorescence emission, and singlet oxygen quantum yield of PpIX are not discussed here.

Previously, Abo-Zeid et al. showed the cytotoxic and genotoxic impact of 5-ALA PDT on the HepG2 cell line [21]. Since we used an in-house-developed laser illumination system along with light fractionation, we did not use any previously published PDT dose for this study. A series of illuminations and pauses, called light fractionation, has been demonstrated to enhance the overall efficacy of the therapy, probably by improving the influx of intracellular oxygen [22,23].

Our in vitro results, however, demonstrated the significant impact of the 5-ALA Only or Light Only control conditions on the metabolic activity, proliferation, and cytotoxicity of the HCC cell lines. Illumination at the 635 nm wavelength induced a significant increase in the metabolic activity of the cell lines while increasing LDH release as well. We can assume the bioactivation of the cells due to an endogenous sensitizer when exposed to light with a wavelength of 635 nm. Similarly, 5-ALA alone decreased the proliferation of cells, highlighting that 5-ALA or its intermediates might affect the cell cycle and proliferation. These results can be explained by the recent findings of Grigalavicius et al., in which they highlighted the potential inhibition of cellular glycolysis by 5-ALA, due to its structural similarity with LDH inhibitors [24]. But we did not observe such effects during in vivo studies, which calls to attention a very crucial point for PDT-associated in vitro studies, i.e., the lack of suitable study models. During in vitro studies, classical 2D cultures are used at normal oxygen levels, with homogenous illumination conditions. Such ideal conditions are rarely available in clinic, where hypoxia and non-uniform light distribution due to tissue pigmentation are commonly observed. We strongly recommend the use of spheroids and 3D culture models along with hypoxic conditions for in vitro studies, employing efficient light delivery devices [25,26].

Taking into consideration the similar levels of enzyme expression, along with the similar levels of intracellular PpIX accumulation in the three cell lines, it seems that the varying sensitivity of the cell lines for 5-ALA PDT is not associated with PpIX levels. There can be other intrinsic factors responsible, like a variation in p53 expression. The activation of p53 is a central factor in the regulation of cellular proliferation, cell death, and the resistance to chemotherapy [16]. p53 signaling is altered in roughly 70% of HCC patients, along with WNT signaling and telomerase promotor, and can affect the sensitivity of a patient to any given therapy [27]. Considering the variation in the p53 state of the HCC cell lines, we can probably explain the variation in PDT-associated sensitivity. Furthermore, our results highlighted that 5-ALA PDT activates necrosis of HuH7 cells, thereby suggesting a p53-mediated necrotic cell death pathway. This is in line with results published by Montero et al., which described ROS-mediated p53-dependent necrotic cell death [28] through cathepsin Q regulation [29], thereby highlighting the role of p53 in regulating non-apoptotic cell death pathways. Our unpublished data highlight the 5-ALA PDT-mediated necrotic cell death of HepG2 cells. Interestingly, Yow et al. demonstrated that the application of 5-ALA PDT on HepG2 causes cell death by apoptosis and the upregulation of p53 [30]. Since we used a lower 5-ALA dose and irradiance than what is reported in this study, we can hypothesize that different PDT doses can induce different cell death pathways, an important parameter for the clinical success of the therapy. Furthermore, Tong et al. hypothesized that p53 affects the sensitization of human fibroblasts for PDT mediated by Photofrin, also known as porfimer sodium [31]. All these studies, thus, highlight the crucial role of this transcription factor in the mechanism of PDT in general, which can vary from cell to cell and between the differences in the PDT protocols. Understanding its role in the efficacy of a therapy could give crucial information about the underlying mechanism and thereby facilitate the clinicians in deciding the good responders to the therapy.

Further, we demonstrated the effect of PDT on primary samples. Previous studies had demonstrated that fibroblast activation causes fibrosis in the HCC tumor microenvironment, which ultimately leads to poor prognosis [32]. Hence, we treated HLMFs with 5-ALA PDT and concluded that PDT does not exhibit any impact on HLMFs in terms of viability and activation. To our knowledge, this is the first study to investigate the impact of PDT on fibroblasts and evaluate the induction of fibrosis. Similarly, this is the first study to present the impact of PDT on primary patient-derived tumor hepatocytes. We under-stand that our tumor hepatocytes are not exclusive to other cellular components of the TME. But to eliminate this possibility, we specifically harvested cells from the core of the tumor section, which anatomopathological analysis later confirmed to be tumours. Additionally, these hepatocytes were cultured in high density to avoid the proliferation of fibroblasts.

By utilizing the conditioned media from 5-ALA PDT treated cells, we tried to mimic the extracellular factors present in the TME, as well as their impact on human immune cells and tumor cells. We demonstrated a possible anti-tumor immune response generated by the secretomes of 5-ALA-PDT-treated HCC cells. However, the immune population activated, and its activation state remains to be studied. This is crucial, as it will define the type of anti-tumor immune response generated by the therapy, memory cell induction, and further initiation of immunogenic cell death. PDT is widely recognized as an inducer of immunogenic cell death [11,33]. Using pheophorbide-a-mediated PDT on HepG2 cells, Tang et al. demonstrated an improved efficacy of the therapy via the activation of an anti-tumor immune response [34]. Such a response can be generated by various factors, notably by tumor exosomes. Recently, Baydoun et al. demonstrated that extracellular vesicles secreted by ovarian cancer cell lines treated with novel folate-coupled PS-mediated PDT could induce CD4+ or CD8+ T cell activation [35]. When tested for cytokine release, we did not detect any change for IL-2, IL-6, IL-10, TGF-β, TNFα, or IFNγ (results not shown). Further, the tumor cytostatic effect generated by conditioned media from 5-ALA PDT is very important in the context of solid tumors like HCC since the effective penetration of light decreases as a function of distance. Hence, it highlights how PDT over the treated region will impact the untreated region of cancer.

In a review article recently published by our team, we discussed various challenges associated with PDT for HCC, namely PS bioavailability, light penetration, availability of intracellular oxygen, liver pigmentation, and PpIX metabolism by non-tumor hepatocytes [36]. All these factors can contribute immensely to the success or failure of 5-ALA PDT, therefore we validated our in vitro findings by using a subcutaneous SCID mice model of HCC treated with an optimal 5-ALA PDT dose. For these in vivo experimentations, we used 200 mg/kg of 5-ALA, which is 10 times to dose commonly used in the clinic, with 18 h of incubation, along with 45 mW/cm^2^ of illumination dose with 2 min of illumination and 2 min of pause. Our unpublished data revealed an effective PpIX accumulation and subsequent fluorescence generation in the tumor but not in non-tumor tissues. Here, we used a subcutaneous model for HCC rather than an orthotopic model, as the light penetration will be highly decreased in the latter. Additionally, the subcutaneous model is in line with our therapeutic strategy for intraoperative PDT for HCC, which is presented in our review [36].

The illumination device used here has been published recently [37] while the protocol used has been standardized by our unpublished study. The 18 h drug-to-light interval is based on our in vitro results, which suggest a higher PpIX accumulation in the HuH7 cells until 18 h. This rationale is also supported by other publications. Egger et al. had previously established the in vivo efficacy of 5-ALA PDT on a subcutaneously implanted rat model of hepatoma, with which they demonstrated that tumor tissues had a higher accumulation of PpIX and for a longer duration, and thus, showed a higher degree of necrosis than the control groups [38]. These findings were later verified by Otake et al., who demonstrated a higher tumor-selective PpIX-based photodynamic diagnosis (PDD) and PDT for a chemically induced HCC [39]. Nishimura et al. demonstrated that the use of 5-ALA-mediated photodynamic diagnosis (PDD) to detect the real-time fluorescence of PpIX in the tumor of all 12 patients [10]. Inoue et al. validated these results with a larger patient group [9]. In the context of other solid tumors of the abdominal cavity, Wagner et al. highlighted the efficacy of temoporfin-mediated PDT for advanced biliary tract carcinoma, a type of liver cancer, for 29 patients in a phase II study [40]. Recently, Quilbé et al. investigated the impact of novel folate-coupled PS-mediated PDT on humanized SCID mice model of pancreatic adenocarcinoma and demonstrated a decrease in the growth of the tumor over time [41]. This, along with our study, highlights the possible application of PDT on other solid tumors.

## 4. Materials and Methods

### 4.1. In Vitro Cell Models

#### 4.1.1. HCC Cell Lines

Three human HCC cell lines were used: HuH7, HepG2, and Hep3B. HuH7 and HepG2 were provided as a courtesy by Prof. Filomena Conti (UPMC, Paris, France). Hep3B was provided as a courtesy by Prof. Jean Dubuisson (CIIL, Lille, France). The cell lines were cultured in RPMI 1640 (Gibco, Waltham, MA, USA) supplemented with 10% (*v*/*v*) decomplemented and filtered fetal calf serum (FCS) (Eurobio, Les Ulis, France) along with 100 units/mL of penicillin, 100 μg/mL of streptomycin (Gibco, Waltham, MA, USA). Cells were maintained at 37 °C, 5% CO_2,_ and 95% humidity. A luciferase-expressing HuH7 cell line (HuH7-Luc), developed and provided as a courtesy by Prof. Antoine Galmiche (Lymphocyte Normal—Pathologique et Cancers Lab, Amiens, France), were cultured in the above-mentioned media along with frequent passage with the neomycin selection gene (Sigma-Aldrich, Saint Louis, MO, USA) at 10 μg/mL for 5 to 7 days.

#### 4.1.2. Tumor Hepatocytes from HCC Patients

Primary tumor hepatocytes were isolated from HCC patient explants and given as a courtesy by Prof. Emmanuel Boleslawski (CHU of Lille, Lille, France). Explicit consent was obtained from each patient prior to surgery and the protocol was validated as part of the hospital tumor banking process (Tumorothèque ALLIANCE CANCER, Centre de Biologie-Pathologie, 59037 Lille CEDEX, France). The explant was washed twice with PBS (Gibco, Waltham, MA, USA) and Betadine 10% (MEDA Pharma GmbH, Wangen-Brüttisellen, Switzerland) before undergoing mechanical shredding, and enzymatic digestion, using collagenase 1A (5 mg/mL, Sigma Aldrich, Saint Louis, MO, USA) and dispase (10 mg/mL, Life Technologies, Carlsbad, NM, USA). This suspension was then incubated for 1 h at 37 °C, with regular shaking. The solution was then centrifuged (800× *g*, 5 min, 20 °C), and the pellet was washed with PBS and centrifuged again (800× *g*, 5 min, 20 °C). Thereafter, the cells were washed with red blood cell lysis buffer (Miltenyi Biotec, Bergisch Gladbach, Germany) for 10 min. Lastly, the cells were centrifuged again (800× *g*, 5 min, 20 °C) to recuperate the pellet in a complete DMEM culture medium (Gibco, Waltham, MA, USA) (10% FCS + Gentamicin), counted, and seeded at 50,000 cells/100 μL of complete media in a white-wall 96-well plate (Corning, Somerville, MA, USA). The cells were used after at least 48 h of culture and treated for different time points.

#### 4.1.3. Liver Myofibroblasts from Healthy Human Donors (HLMFs)

HLMFs from three healthy donors were provided by IHU ICAN- Liver Biology liver platform with due consent from the donors. The cells were maintained in DMEM (Gibco, Waltham, MA, USA) supplemented with 10% (*v*/*v*) of decomplemented and filtered FCS (Eurobio, Les Ulis, France), 1% sodium pyruvate (Gibco, Waltham, MA, USA), 100 units/mL of penicillin, 100 μg/mL of streptomycin (Gibco, Waltham, MA, USA) and 1% Zell Shield (Minerva BioLabs, GmBH, Berlin, Germany). The cells were cultured at 37 °C, 5% CO_2_, and 95% humidity and the media was changed every 48 h.

#### 4.1.4. Peripheral Blood Mononuclear Cells from Human Healthy Donor (PBMCs)

Human peripheral blood mononuclear samples were collected from healthy adult donors with informed consent in accordance with the approval of the EFS board (Etablissement Français du Sang). PBMCs were isolated by density gradient centrifugation of the blood using a lymphocyte separation medium (Eurobio, Les Ullis, France) and 50 mL Leucosep tubes (Greiner Bio One, Courtaboeuf, France). In a 96-well plate (Corning, Somerville, MA, USA), 100,000 PBMCs were either stimulated or not stimulated by 1 µg/mL of phytohaemagglutinin (PHA) (Sigma-Aldrich, Saint Louis, MO, USA) cultured for 5 days with 50:50 of conditioned media (see PDT section) and ML10 media (RPMI 1640 medium, sodium pyruvate (1 mM), nonessential amino acids MEM 1×, HEPES (25 mM), 2-mercaptoethanol(50 μM), gentamicin (10 μg/mL)) (Thermo Fisher Scientific, Waltham, MA, USA), and 10% FCS (Gibco, Waltham, MA, USA), to measure their viability and cellular proliferation at regular time points.

### 4.2. In Vivo SCID Mice Model of HCC

All procedures were approved by the local ethical committee of the Institut Pasteur de Lille and performed with the required permission of the national governing ethical board (approval number 2019041015585930), and the mice received humane care. All male mice, aged 6 to 8 weeks, were used and kept in pressurized and individually ventilated cages with a regular mice diet of 10% animal fat. Anesthetized SCID mice were subcutaneously injected with 10 million HuH7-Luc cells in 100 μL Matrigel (Corning, Somerville, MA, USA) and observed for tumor growth by using intraperitoneal injection of 100 μL of D- luciferin (30 mg/mL, Perkin Elmer, Waltham, MA, USA) over an IVIS LUMINA XR reader (Caliper LifeSciences, Hopkinton, MA, USA), and analyzed under Living Image 4.1 software (CaliperLife Sciences, Hopkinton, MA, USA). Results were obtained after spectral unmixing according to the manufacturer’s instructions and expressed in normalized bioluminescence.

### 4.3. Photodynamic Therapy

For in vitro assays, 5000 cells per 100 μL of media for each cell line were seeded in a 96-well plate (Corning, Somerville, MA, USA) in triplicate and incubated for 4 h with or without 5-ALA (Sigma-Aldrich, Saint Louis, MO, USA) at various concentrations (0.1 mM to 0.6 mM) following irradiance using a laser set-up at 635 nm with varying dosage (0 J/cm^2^ to 3.6 J/cm^2^) [37,42]. Cellular viability was assessed at 24 h post-illumination. The power output of the laser was set to 1 mW/cm^2^ with a fractionated illumination consisting of 2 min of illumination and 2 min of pause as a standard illumination protocol while keeping all the non-illuminated conditions next to the illumination device wrapped in aluminum foil, in order to normalize any impact on the cells being out of the incubator during this period. All the experiments were performed in dark conditions. Separate 24-well culture plates (Dutscher, France) were used to obtain microscopic images of cells treated with 5-ALA PDT at 24 h post-illumination. For the recuperation of conditioned media, cells were cultured in T75 culture flasks (Sarstedt, Nümbrecht, Germany) till 70% confluence with 10 mL complete media, followed by 5-ALA PDT at optimal dose and recuperation of media 24 h post-illumination.

For in vivo experimentations, mice were divided into four groups: Non-Treated, 5-ALA Only, Light Only, and 5-ALA PDT Treated. For the 5-ALA Only and 5-ALA PDT condition, 100 μL of 5-ALA (Sigma-Aldrich, Saint Louis, MO, USA) solution dissolved in distilled water was injected intraperitoneally at a dose of 200 mg/kg per mice 18 h before illumination. For the Light Only and 5-ALA PDT group, the mice were anesthetized and subjected to illumination using laser-set at 635 nm with 32.4 J/cm^2^ [37,42]. The power output was set to 12 mW/cm^2^ with 2 min of illumination followed by 2 min of pause, as a standard protocol. Images were then analyzed under Living Image 4.1 software (CaliperLife Sciences, Hopkinton, MA, USA), and the results were obtained after spectral unmixing according to the manufacturer’s instructions. Results were expressed in normalized bioluminescence. Normalized bioluminescence = Bioluminescence of the sample at time point/Average bioluminescence of Non-Treated control at day 0 (day of PDT).

### 4.4. Viability Assay

The cells were cultured in a white-wall 96-well Costar plates (Corning, Somerville, MA, USA) as required cell density in order to triplicate each condition (Non-Treated, Light Only, 5-ALA Only, PDT Treated). 100 µL of Celltiter-Glo mix (Promega, Madison, WI, USA), which determines the cellular viability in a culture based on the quantification of the ATP present, was prepared according to the manufacturer’s instructions, was added to each well and incubated at room temperature for 10 min in the dark. Luminescence reading was performed using the Luminometer centro LB960 (Berthold Technologies, Oak Ridge, TN, USA) running MikroWin software Version 4.41 (Mikrotek Laborsysteme GmbH, Overath, Germany). Results were expressed in relative luminescence units (RLU) or normalized RLU. Normalized RLU = RLU of the sample/Average RLU of Non-Treated control.

### 4.5. Flow Cytometry

80,000 HuH7 cells were seeded in a 6-well plate (Dutscher, Courtaboeuf, France) with 2 mL complete growth media. After overnight incubation, the cells were subjected to an in vitro 5-ALA PDT dose corresponding to the HuH7 cell line, as described above. Cells treated with heat (at 75 °C for 15 min) and staurosporine (Sigma-Aldrich, Saint Louis, MO, USA) (10 µM for 18 h with media containing 1% SVF) were used as positive controls for necrosis and apoptosis, respectively. The cell death was analyzed using an Annexin V-FITC kit (Miltenyi Biotec, Bergisch Gladbach, Germany). The cells were trypsinized and recuperated at 1 h, 4 h, and 24 h post-illumination, processed according to manufacturer’s instructions, and examined using a BD Canto II flow cytometer (Becton Dikinson, Franklin Lakes, NJ, USA) at the BioImaging Center Lille (BICeL) platform. The data were analyzed with FlowJo software (FlowJo, Ashland, OH, USA), and the results were expressed in the percentage of cells of the parent population for each condition.

### 4.6. Fluorimetry

5000 cells per 100 μL of media for each cell line were seeded in a black wall 96-well plate (Corning, Somerville, MA, USA) in triplicate. 24 h after seeding, 5-ALA was added to the wells with a final concentration of 0.6 mM for varying incubation time points (0, 1, 2, 4, 8, 18, 24 h). Throughout the experiment, the cells remained in total darkness. At the end of the incubation period, the media was recovered to measure the extracellular levels of PpIX fluorescence, while fresh media was added to the cells to measure the intracellular levels. Non-Treated controls (without 5-ALA) were established. PpIX fluorescence was measured by an excitation wavelength of 400 ± 10 nm and an emission wavelength of 650 nm using FLUOstar OPTIMA (BMG Labtech, Champigny sur Marne, France) running OPTIMA software version 2.20R2 (BMG Labtech, Champigny sur Marne, France). The results were expressed in relative fluorescence units (RFU).

### 4.7. Proliferation Assay

Proliferation assays were set up in a round-bottom 96-well plate (Corning, Somerville, MA, USA) in triplicate and measured by adding radioactive [^3^H] thymidine (1µCi/well) (PerkinElmer, Courtaboeuf, France) to each well 18 h before harvesting. At the end of the culture, the cells were harvested on a glass fiber filter (PerkinElmer, Courtaboeuf, France) using a Tomtec harvester (Wallac, Turku, Finland), then sealed in a sample bag (PerkinElmer, Courtaboeuf, France) with scintillation liquid (Beckman Coulter, Brea, CA, USA). Radioactive thymidine was measured by scintillation counting using a β-counter (1450 Trilux, Wallac, Finland) [41]. Proliferation was estimated in count per minute (CPM) and expressed in CPM or normalized CPM. Normalized CPM = CPM of the sample/Average CPM of Non-Treated control.

### 4.8. Culture of Cancer Cell Lines with Conditioned Media

5000 cells from HCC cell lines were seeded in a 96-well plate (Corning, Somerville, MA, USA) with 50:50 of conditioned media of the respective cell line and fresh media to monitor their cellular proliferation and viability over a course of 5 days.

### 4.9. RNA Extraction and RT-qPCR

Total RNA extraction from different samples was done using the method previously described [41] and quantified using a UV spectrophotometer. The RT-PCR reactions were performed for selected genes (Table 1) according to the manufacturer’s instructions using 2 × MESAGREEN qPCR MasterMix Plus for the SYBR 258 Assay (Eurogentech, Liège, Belgium) in a 96-well qPCR plate (Sarstedt, Nümbrecht, Germany), and the Mx3005P^TM^ sequence detection system (Agilent Technologies, Santa Clara, CA, USA). A quantitative analysis was made based on the cycle threshold (Ct) value for each well and calculated using MxPro software (Agilent Technologies, Santa Clara, CA, USA). The results were normalized using the mean Ct of the three housekeeping genes (Ct HKG) (Table 1) and the data are represented as dCt or as fold differences using the 2^−ddCt^ method, where dCt = Ct target gene − Ct HKG.

### 4.10. Cytotoxicity Assay

5000 cells of each of the cell lines were seeded in a 96-well plate (Corning, Somerville, MA, USA) and treated with the above-mentioned in vitro 5-ALA PDT protocol, followed by recuperation of the media at 24 h post-illumination. Cells treated with 2% Triton X-100 (Sigma-Aldrich, Saint-Louis, MO, USA) for 2 h were considered as the positive control. The media was used immediately to analyze cytotoxicity by using a Cytotoxicity Detection (LDH) kit (Roche, Sigma Aldrich, Saint Louis, MO, USA) according to the manufacturer’s instructions. The measurements were done using a UV spectrophotometer (Multiskan RC, Thermo Fischer Scientific, Waltham, MA, USA) at 492 nm and was powered by Ascent™Software v2.06 (Thermo Fisher Scientific, Waltham, MA, USA). The cytotoxicity was represented as Relative Cytotoxicity (%) = ((Sample value − Non-Treated Control)/(Positive Control − Non-Treated Control)) × 100.

### 4.11. ELISA of Collagen I

HLMFs were seeded in a 6-well plate (Corning, Somerville, MA, USA) with a seeding density of 5000 cells/100 µL, treated with 5-ALA PDT, and the media was recuperated 24 h post-illumination. Collagen I levels were analyzed for the three controls and the test condition using a Quidel MicroVue CICP EIA kit for ELISA (Quidel, San Diego, CA, USA) and the provided standard range following the manufacturer’s instructions. The measurements were obtained using a UV spectrophotometer (Multiskan RC, Thermo Fisher Scientific, Waltham, MA, USA) at 405 nm powered by Ascent™Software v2.06 (Thermo Fisher Scientific, Waltham, MA, USA). The concentration was represented in ng/mL.

### 4.12. Statistical Analysis

All data were analyzed using the statistical package GraphPad Prism for Windows 3.0.1 (GraphPad, San Diego, CA, USA). All quoted *p*-values are two-sided, with *p* ≤ 0.05 (*), *p* ≤ 0.01 (**), *p* ≤ 0.001 (***), and *p* ≤ 0.0001 (****) being considered statistically significant for the first and highly significant for the other.

## 5. Conclusions

In summary, the present study demonstrates that PDT, using the 5-ALA pro-drug, can be an effective treatment for HCC, as proven by our in vitro and in vivo experiments. The therapy does not induce any effect on liver myofibroblasts. We also demonstrated an immune-stimulatory impact by the secretomes of 5-ALA PDT treated cells, which can also inhibit the proliferation of cancer cells. Nevertheless, 5-ALA-mediated PDT needs to be tested on in vivo mouse models in the presence of an immune system in order to analyze its clinical functionality and the development of future preclinical and clinical studies.

One of the most important applications of our study is the development of 5-ALA PDT in combination with a partial hepatectomy, where it could not only work as an anti-tumor therapy but also as a visual aid during surgery while activating an anti-tumor immune response. We have comprehensively discussed the strategy to use PDT as an intraoperative procedure for the treatment of HCC in a review [36]. Additionally, the immune-stimulatory effect of 5-ALA PDT can help enhance the effect of various immunotherapies currently in trial for HCC. Our study is a crucial key for the development of PDT for HCC treatment and for laying the groundwork for combining intraoperative PDT with immune checkpoint inhibitors or adoptive-T-cell-transfer based immunotherapies.

## Figures and Tables

**Figure 1 ijms-24-10426-f001:**
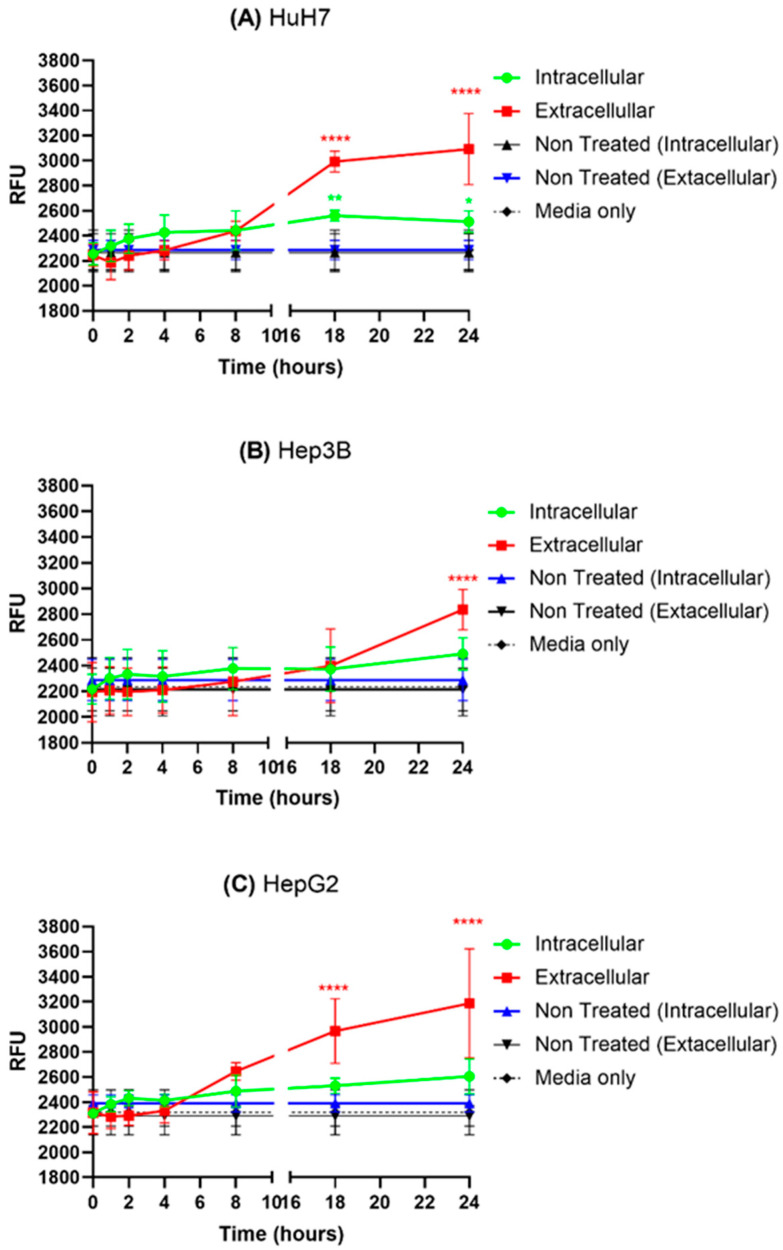
Fluorimetry-based protoporphyrin IX (PpIX) quantification of three HCC cell lines treated by 0.6 mM of 5-ALA for different incubation periods (0, 1, 2, 4, 8, 18, and 24 h). Extracellular PpIX levels were determined by fluorometric analysis of the conditioned media, while intracellular levels were measured by cells in fresh media. The values are represented in relative fluorescence units (RFU). A one-way ANOVA test was performed, with *p* ≤ 0.05 (*), *p* ≤ 0.01 (**), *p* ≤ 0.0001 (****) being considered statistically significant for the first and highly significant for the other (*n* = 2).

**Figure 2 ijms-24-10426-f002:**
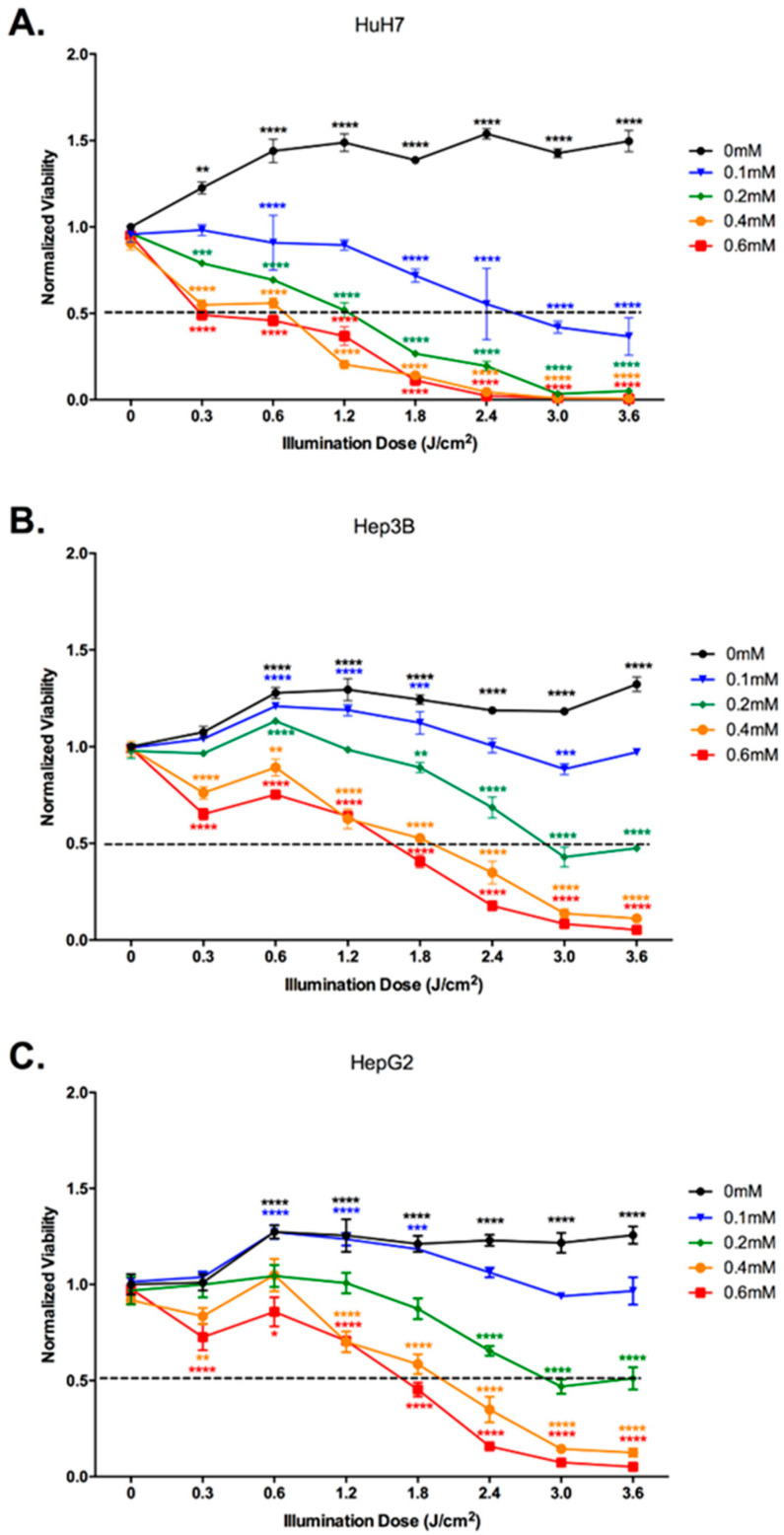
Viability analysis of HCC cell lines (**A**) HuH7, (**B**) Hep3B, and (**C**) HepG2 treated with different concentrations of 5-ALA and illumination doses at 24 h post illumination. The cells were Non-Treated or subjected to different 5-ALA concentrations (at 0 J/cm^2^, from 0 mM to 0.5 mM), or different illumination doses (at 0 mM of 5-ALA and up to 3.6 J/cm^2^) and illumination time variations (5 min, 10 min, 20 min, 30 min, 40 min, 50 min, 60 min), or were PDT-treated (within the same dose range) using a laser set-up at 635 nm with an irradiance rate of 1 mW/cm^2^. The viability readings were then normalized by the Non-Treated Control. A two-way ANOVA test was performed with *p* ≤ 0.01 (**), *p* ≤ 0.001 (***), and *p* ≤ 0.0001 (****) being considered statistically significant for the first and highly significant for the others (*n* = 3).

**Figure 3 ijms-24-10426-f003:**
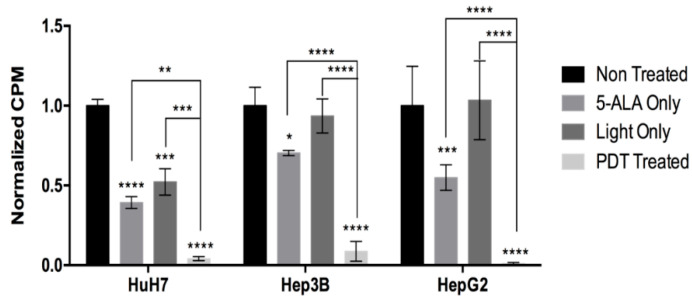
Proliferation assay of HCC cell lines treated by their respective IC50 dose, 24 h post illumination. The cells were subjected to no treatment (Non-Treated), 0.6 mM of 5-ALA only (5-ALA Only), 0.6 J/cm^2^ or 1.8 J/cm^2^ of illumination (Light Only), or were PDT Treated. The counts per minute (CPM) readings were then normalized by the Non-Treated control. A two-way ANOVA test was performed with *p* ≤ 0.05 (*), *p* ≤ 0.01 (**), *p* ≤ 0.001 (***), and *p* ≤ 0.0001 (****) being considered statistically significant for the first and highly significant for the others (*n* = 3).

**Figure 4 ijms-24-10426-f004:**
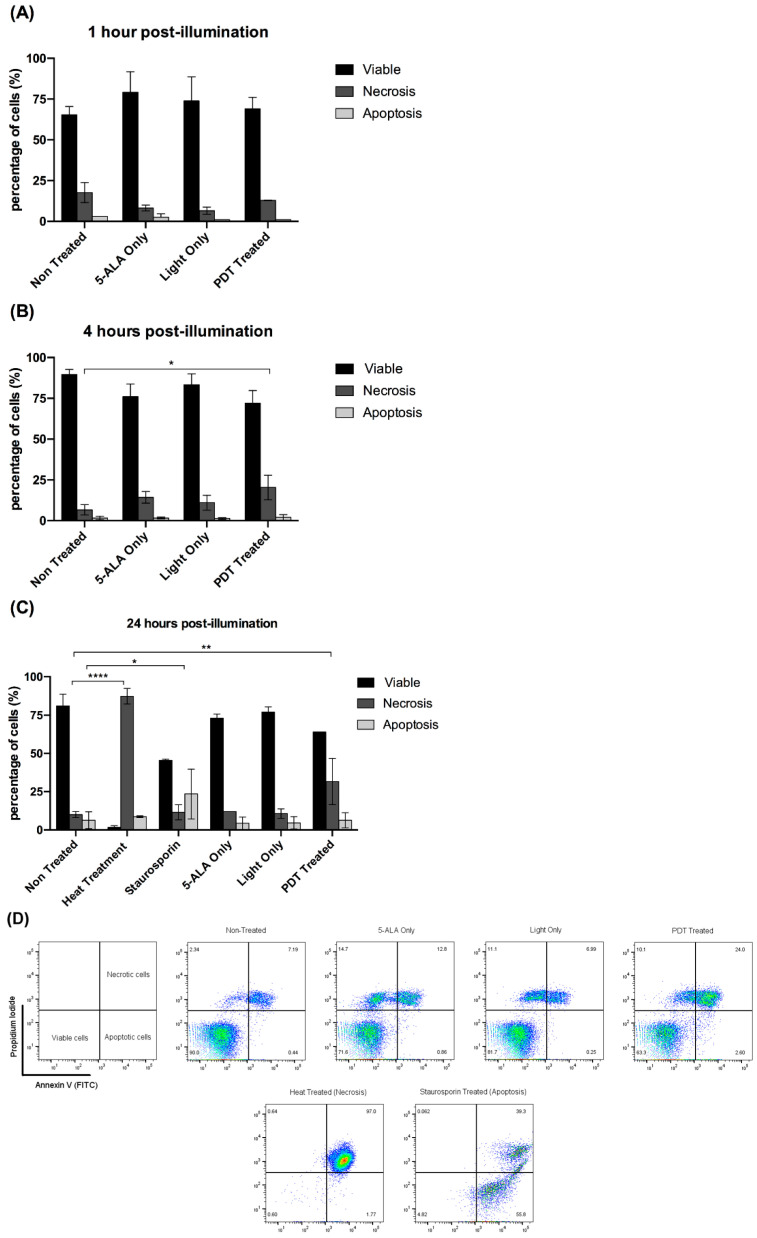
Annexin-V/PI-based flow cytometry for analysis of the type of cell death induced by 5-ALA PDT for HuH7 cell line at (**A**) 1 h, (**B**) 4 h, and (**C**) 24 h post-illumination. The values are expressed as a percentage of the parent population, where the values were compared with the Non-Treated condition. Heat treatment (75 °C for 15 min) and staurosporine (10 µM for 18 h) were used as positive controls for necrosis and apoptosis, respectively. A two-way ANOVA test was performed, with *p* ≤ 0.05 (*), *p* ≤ 0.01 (**), and *p* ≤ 0.0001 (****) being considered statistically significant for the first and highly significant for the others (*n* = 3). (**D**) Representative flow cytometry plots showing the frequency of viable, necrotic, and apoptotic HuH7 cells, when subjected to no treatment, 5-ALA Only (0.6 mM for 4 h), Light Only (0.6 J/cm^2^), PDT Treated (0.6 mM 5-ALA for 4 h with 0.6 J/cm^2^ light dose), Heat Treated and Staurosporine Treated. Representative pseudocolor dot plot.

**Figure 5 ijms-24-10426-f005:**
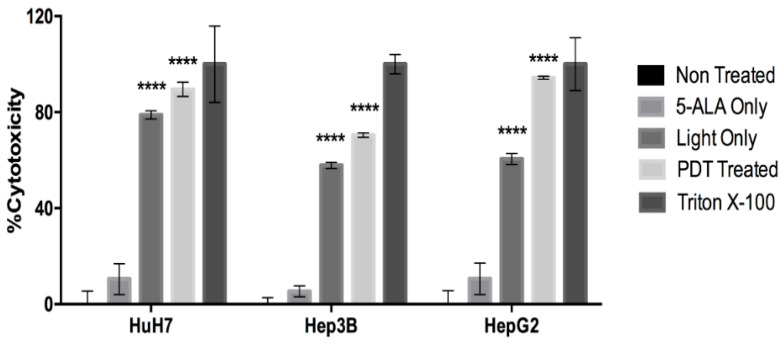
Lactate dehydrogenase (LDH- release-based cytotoxicity was analyzed for HCC cell lines treated by 5-ALA PDT at 0 h and 24 h post-illumination. The values are expressed as % Cytotoxicity, where the values were compared with those of the Non-Treated and positive control (treated by Triton-X). The percent cytotoxicity was determined by the formula described earlier. A two-way ANOVA test was performed, with *p* ≤ 0.00001 (****) being considered statistically significant for the first and highly significant for the others (*n* = 3).

**Figure 6 ijms-24-10426-f006:**
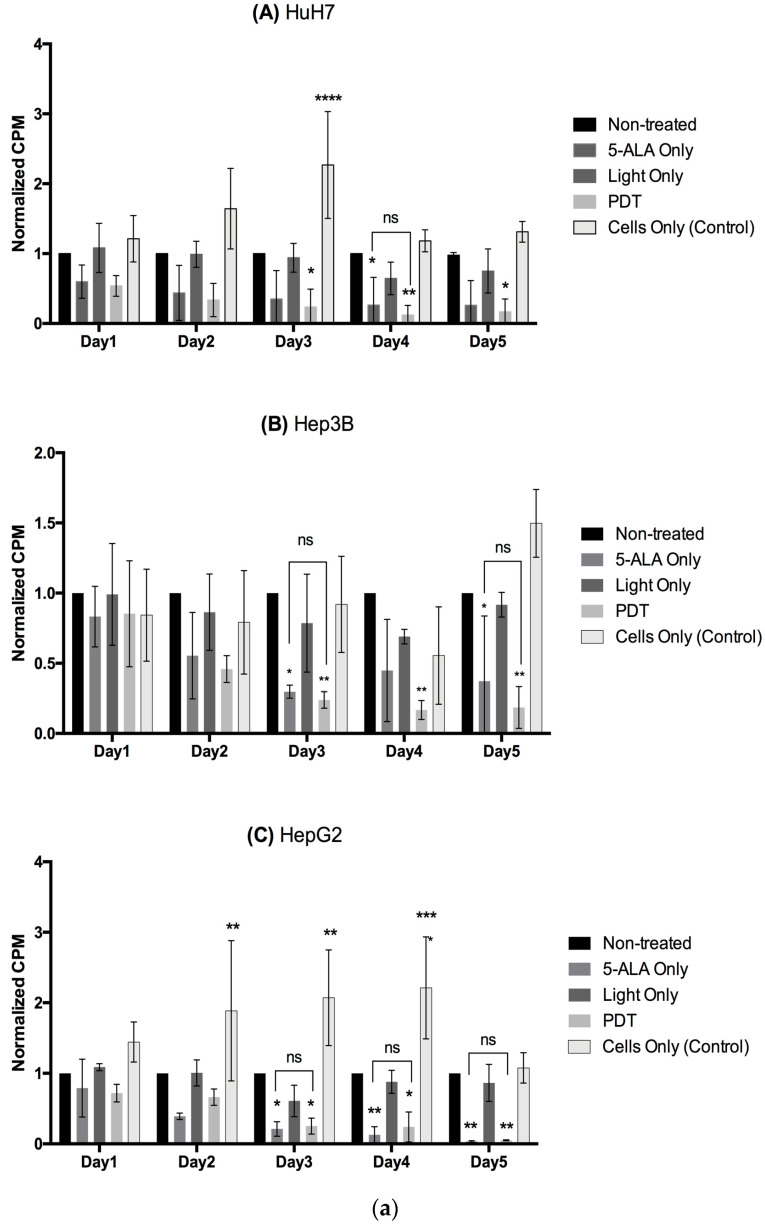
(**a**) Impact of conditioned media on the proliferation of cancer cells. The data are represented in counts per minute (CPM) normalized with Non-Treated control. A two-way ANOVA test was performed, with ns considered as non-significant, *p* ≤ 0.05 (*), *p* ≤ 0.01 (**), *p* ≤ 0.001 (***), and *p* ≤ 0.0001 (****) being considered statistically significant for the first and highly significant for the others (*n* = 3). (**b**) Impact of conditioned media on the viability of cancer cells. The data are represented in viability normalized with Non-Treated control. A two-way ANOVA test was performed with *p* ≤ 0.05 (*), *p* ≤ 0.01 (**), *p* ≤ 0.001 (***), and *p* ≤ 0.0001 (****) being considered statistically sig-nificant for the first and highly significant for the others (*n* = 3).

**Figure 7 ijms-24-10426-f007:**
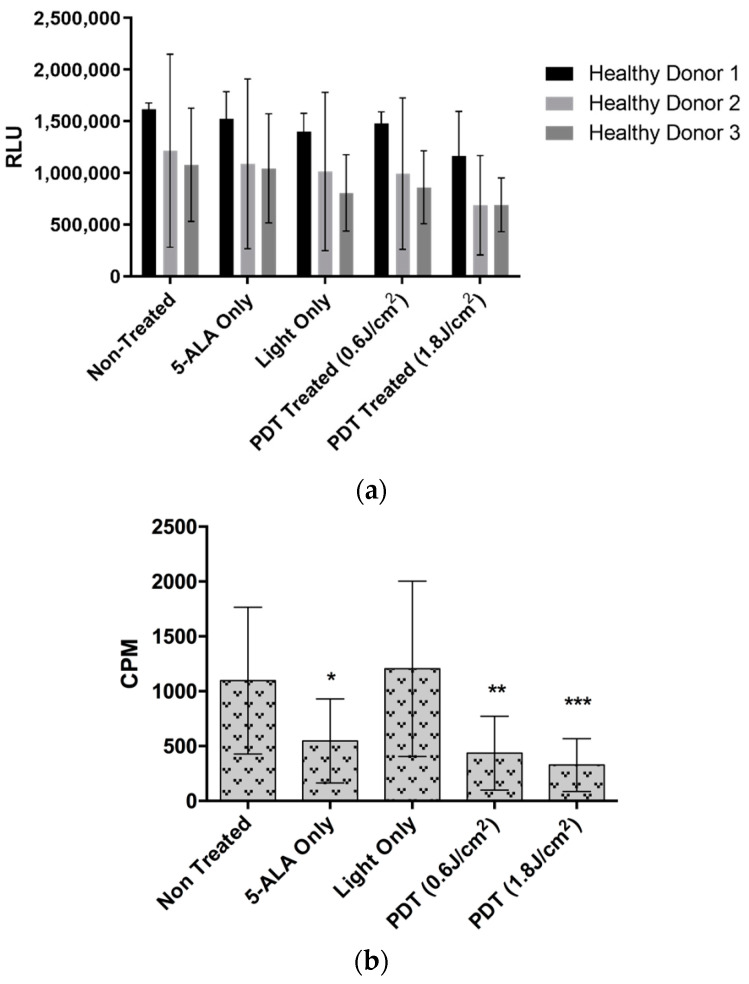
(**a**) Viability analysis of donor healthy liver myofibroblasts (HLMFs) treated with two doses of 5-ALA PDT (0.6 mM of 5-ALA with 0.6 J/cm^2^ and 1.8 J/cm^2^ of illumination dose) at 24 h post illumination. The values are expressed as relative luminescence units (RLU). A two-way ANOVA test was performed (*n* = 3). (**b**) Cellular proliferation analysis of primary healthy liver myofibroblasts (HLMFs) from three different donors treated with two doses of 5-ALA PDT (0.6 mM of 5-ALA with 0.6 J/cm^2^ and 1.8 J/cm^2^ of illumination dose) at 24 h post-illumination. The values are expressed as RLU. A two-way ANOVA test was performed, with *p* ≤ 0.05 (*), *p* ≤ 0.001 (**) and *p* ≤ 0.0001 (***) being considered statistically significant or highly significant, respectively. (*n* = 3).

**Figure 8 ijms-24-10426-f008:**
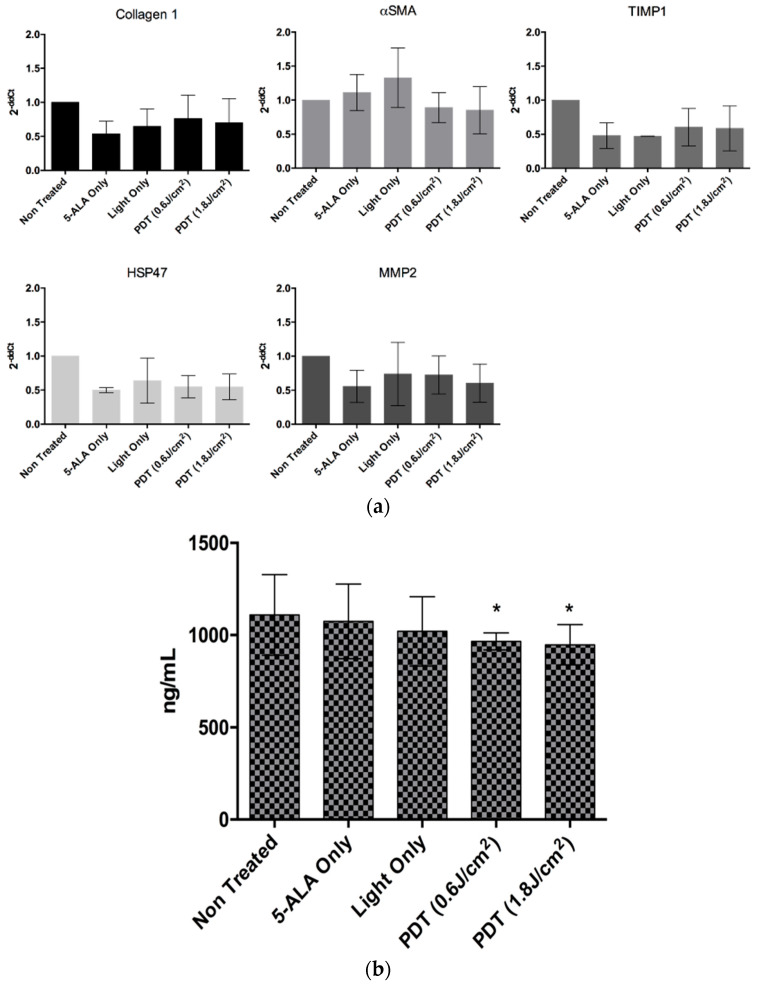
(**a**) Analysis of fibrosis of donor healthy liver myofibroblasts induced by 5-ALA PDT at two doses, 24 h post-illumination, using RT-qPCR-based gene expression analysis of collagen 1, α-smooth muscle actin (αSMA), tissue inhibitor of metalloproteinase 1 (TIMP1), HSP47, and matrix metallopeptidase 2 (MMP2) for different conditions, 24 h post-illumination. A two-way ANOVA test was performed (*n* = 3). (**b**) Analysis of fibrosis of donor healthy liver myofibroblasts induced by 5-ALA PDT at two doses, 24 h post-illumination, by ELISA-based analysis of collagen I secretion. A two-way ANVOVA test was performed, with *p* ≤ 0.05 (*), being considered statistically significant or highly significant. (*n* = 3).

**Figure 9 ijms-24-10426-f009:**
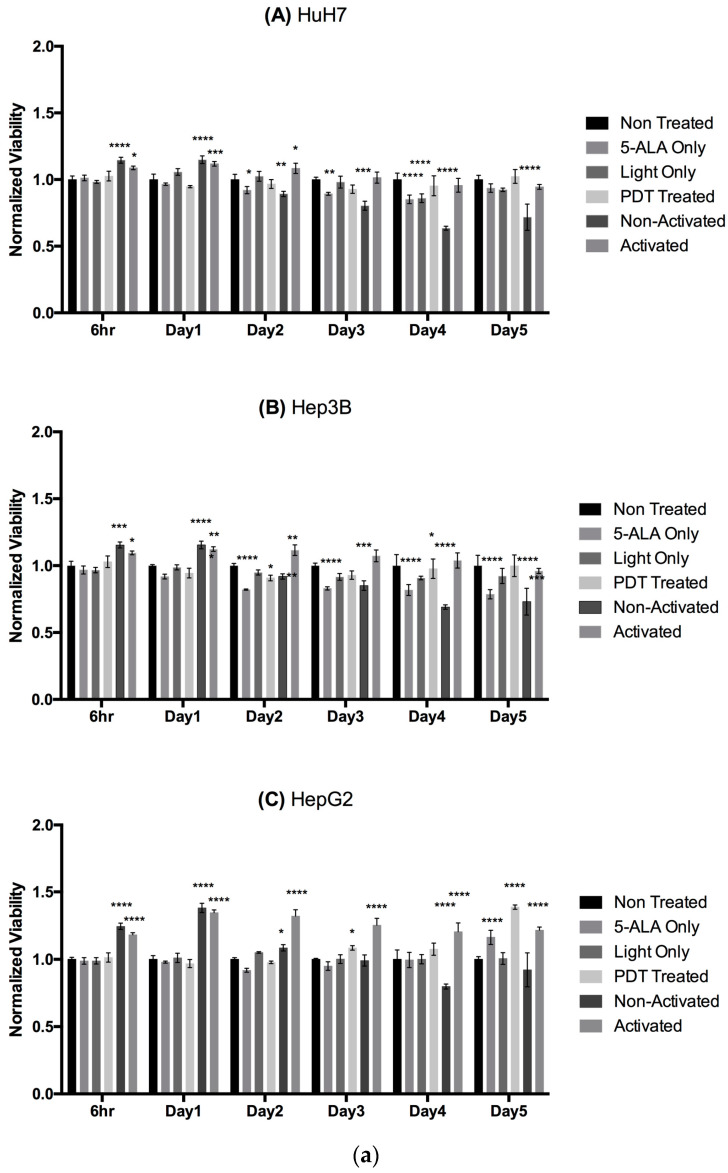
(**a**) Impact of conditioned media from HCC cell lines, (**A**) HuH7, (**B**) Hep3B, and (**C**) HepG2, on the viability of activated human PBMCs. The data are represented in viability normalized with Non-Treated control. A two-way ANOVA test was performed, with *p* ≤ 0.05 (*), *p* ≤ 0.01 (**), *p* ≤ 0.001 (***) and *p* ≤ 0.0001 (****) being considered statistically significant or highly significant, respectively. (**b**) Impact of conditioned media from HCC cell lines, (**A**) HuH7, (**B**) Hep3B, and (**C**) HepG2, on the proliferation of activated human PBMCs. The data are represented in counts per minute (CPM) normalized with Non-Treated control. A two-way ANOVA test was performed, with *p* ≤ 0.05 (*), *p* ≤ 0.01 (**), and *p* ≤ 0.001 (***) and *p* ≤ 0.0001 (****) being considered statistically significant for the first and highly significant for the others (*n* = 3).

**Figure 10 ijms-24-10426-f010:**
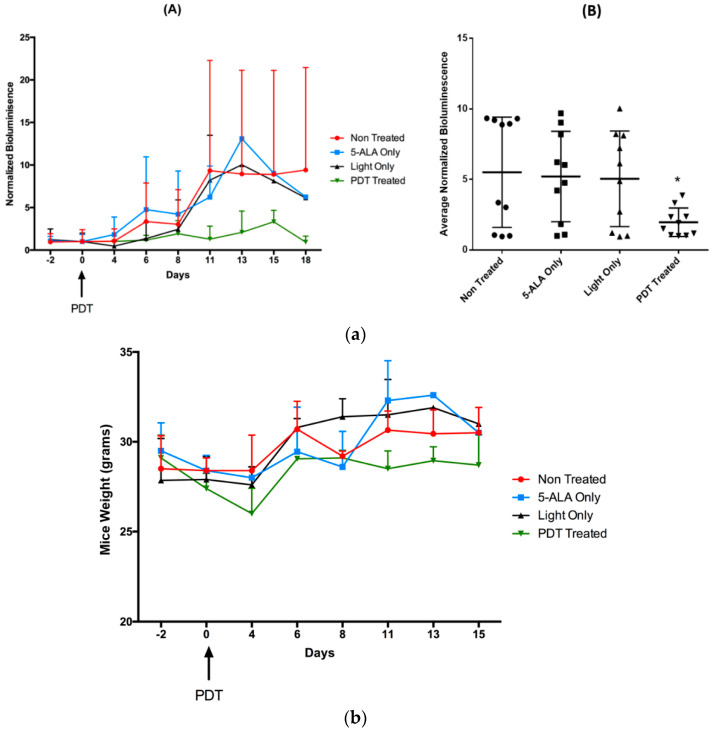
(**a**) (**A**) Normalized bioluminescence of SCID mice with humanized HCC tumor, treated with or without 5-ALA PDT, over the period from 2 days prior to illumination to 15 days post-illumination (*n* = 2) (**B**) Average normalized bioluminescence of different mice groups starting from day 0 to day 20 post-illumination. A two-way ANOVA test was performed, with *p* ≤ 0.05 (*) being considered statistically significant (*n* = 2). (**b**). Average of mice weight of different groups over the period from 2 days prior to illumination to 15 days post-illumination. (*n* = 2). For all the graphs, the circle corresponds to non-treated conditiond, the squares to treatment with 5-ALA only, the upward triangle to treatment with light only and the downward triangle to treatment with PDT.

**Figure 11 ijms-24-10426-f011:**
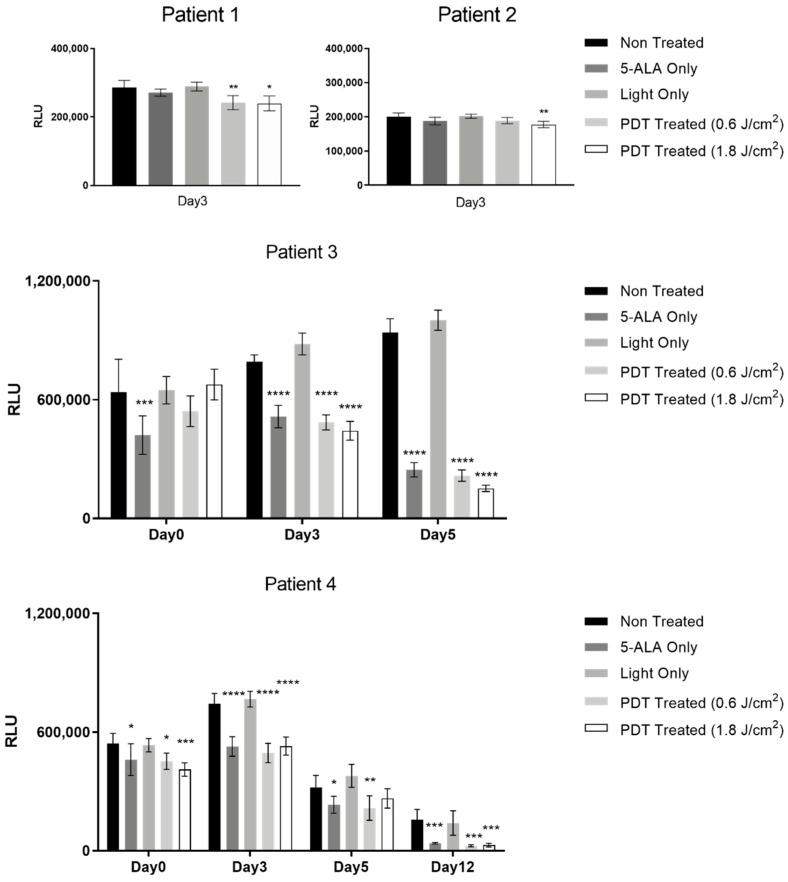
Viability analysis of tumor hepatocytes from four different HCC patients treated with two different doses of 5-ALA PDT (0.6 mM of 5-ALA with 0.6 J/cm^2^ and 1.8 J/cm^2^ of illumination dose), at different time points post-illumination. The values are expressed as relative luminescence units (RLU). A two-way ANOVA test was performed, with *p* ≤ 0.05 (*), *p* ≤ 0.01 (**), *p* ≤ 0.001 (***), and *p* ≤ 0.0001 (****) being considered statistically significant for the first and highly significant for the others.

**Table 1 ijms-24-10426-t001:** List of primers with their sequences in 5′ to 3′ direction.

Gene Name	Forward Sequence	Reverse Sequence
Collagen-1	CCTCAAGGGCTCCAACGAG	TCAATCACTGTCTTGCCCCA
Alpha-Smooth Muscle Actin (α-SMA)	TGAAGAGCATCCCACCCT	ACGAAGGAATAGCCACGC
Tissue Inhibitor of Metalloproteinases 1 (TIMP1)	CCTGTTGTTGCTGTGGCTGA	GGTATAAGGTGGTCTGGTTGACTTC
Heat Shock Protein 47 (HSP47)	TGAAGATCTGGATGGGGAAG	CTTGTCAATGGCCTCAGTCA
Matrix Metalloproteinase 2 (MMP2)	ACGACCGCGACAAGAAGTAT	ATTTGTTGCCCAGGAAAGTG
Beta-Actin	CACGGCATCGTCACCAACT	AGCCACACGCAGCTCATTG
Glyceraldehyde 3-Phosphate Dehydrogenase (GAPDH)	GCCAAGGTCATCCATGACAACTTTGG	GCCTGCTTCACCACCTTCTTGATGTC
Hypoxanthine Phosphoribosyltransferase (HPRT)	CCCTGGCGTCGTGATTAG	ATGGCCTCCCATCTCCTT

## Data Availability

Data are available upon request to the corresponding author.

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
