# Peer review of "An Efficient 5-Aminolevulinic Acid Photodynamic Therapy Treatment for Human Hepatocellular Carcinoma"

_ijms, 2023, doi:10.3390/ijms241310426_

Round 1

Reviewer 1 Report

The work presented by Abhishek Kumar et al studied PDT effect in human hepatocellular carcinoma. This study investigated the effect of 5-ALA on hepatocellular carcinoma in various aspects using the previously well-known 5-ALA. Although the possibility of PDT approached from various aspects has been confirmed, there are some unsatisfactory points.

It is recommended to refine it and publish it through a minor revision process.

1.     It would be nice if an explanation of why the conditions (laser intensity and dose) of PDT that show cytotoxicity (IC50) depending on the cell line work differently would be presented.

2.     In the description of Figure 2, information on the wavelength range of the irradiated light and the total irradiation time is missing.

3.     Comparing the results of Figures 2 and 3, the results of non-treat and light only show a difference. It seems to increase in Figure 2, and it seems to decrease or similar results in Figure 3. What is the reason?

4.     As with the previous figures, it would have been nice to show at least the result of Figure 4 at least a fluorescence image using PI for understanding and credibility of the paper. Is there a reason why you completely excluded the image? In general, most PDT-related research papers claim the reliability of the results through at least one imaging result.

5.     In Figure 5, toxicity is increased even under light only conditions, unlike Figure 2, what difference is there?

6.     The figure2d mentioned in line 136 of the text cannot be found. It looks like it's probably missing, and needs to be checked.

7.     The description of Figure 8 seems to be omitted from the text. Verification is required.

8.     In Figure 11, is there a reason for performing different time conditions for each patient? It would have been a better format for presenting results if all conditions were the same for each patient.

Author Response

Dear reviewer 1,

The point by point answer is in a PDF file.

Regards

Reviewer 2 Report

The manuscript “An Efficient 5-Aminolevulinic Acid Photodynamic Therapy Treatment for Human Hepatocellular Carcinoma” illustrates the effects of a very well-know photosensitizer, 5-Aminolevulinic Acid, on three HCC cell lines upon irradiation. The bibliography is complete, but the narration doesn’t always flows able to capture the interest of the reader. In my opinion, the good quality of this manuscript calls for publication with only minor revisions. The more slightly critical points in the paper are:

-        The authors should revised the text to make reading more fluid and easy. They show and list a lot of data, without paying due attention and appreciation to them. Probably put together “Results and Discussion” could help.

-          The authors should revise the text to eliminate grammar errors and not correct phrases (some as an example are listed below)

-          The authors should better discuss the advantages of using 5-ALA under illumination with respect to 5-ALA alone, because it is not always clear.

Minor points:

-          Some typos remain in the text, separation in the word by “-“

-          Line 105: the authors should better explain what they mean with photoactivation of cells.

-          Line 108: the comment on HuH7 is too general

-          Line 117: authors should eliminate the comma

-          Line 118: it is an example for a more general comment. Authors should explain better, adding only some single phrase to make an inexpert reader able to comprehend their considerations and results. For example, they can add a definition for p53 state or why they follow some pathway to identify the cell death pathway. They explain these information in the paragraph “Discussion” but, probably it may be better to change the order of some part of the text.

-          Figures: the authors should consider to do color figures in order to make more clear the conditions and the results

-          Line 136: which is figure 2(d) cited in the text? I didn’t find any images obtained with a fluorescence microscope

-          In the text the panels are indicate with lower case letters, whereas in the Figures with capitol letters. Authors should align notations.

-          Lines 173-183: in this paragraph, as in others, authors should use more fluid and clear phrases, to facilitate reading the text and interest the reader.

-          Line 200-203: check this sentence (grammar)

-          Lines 208-210: check this sentence (grammar and clarity)

-          Check in the text citations for Figure 8, because I think that there is no one.

-          Line 249: check “maxes”

-          Line 252-253: authors should make more clear and fluid their sentences

-          Line 274-275: authors should better explain. They seem to contradict themselves: do in vitro results demonstrate toxicity induced by PDT or not?

-          Figure 10: The error barrs associated to “no treated samples” are too high to allow a comparison with other conditions. Moreover, the authors should better explain the results shown in Figure 10.

-          Line 315: the authors should correct in “singlet oxygen quantum yield “

The authors should revise the text to eliminate grammar errors and not correct phrases (some as an example are listed in the comments)

Author Response

Dear Reviewer,

the point by point response is in the PDF file.

Regards

Round 2

Reviewer 1 Report

The work presented by Abhishek Kumar et al studied PDT effect in human hepatocellular carcinoma. This study investigated the effect of 5-ALA on hepatocellular carcinoma in various aspects using the previously well-known 5-ALA. The author seems to have carried out sufficient review and effort for the corrections requested above. If a little more bioimaging data is added in future research, it will likely become a more plausible research paper.

Author Response

Thank you